# Polyelectrolyte Microcapsules: An Extended Release System for the Antiarrhythmic Complex of Allapinin with Glycyrrhizic Acid Salt

**DOI:** 10.3390/ijms25052652

**Published:** 2024-02-24

**Authors:** Shavkat I. Salikhov, Egor V. Musin, Aleksandr L. Kim, Yulia I. Oshchepkova, Sergey A. Tikhonenko

**Affiliations:** 1Institute of Bioorganic Chemistry Named after A.Sadykov Academy of Sciences of the Republic of Uzbekistan, M. Ulugbek Str., 83, Tashkent 100125, Uzbekistanjoshepkova05@rambler.ru (Y.I.O.); 2Institute of Theoretical and Experimental Biophysics Russian Academy of Science, Institutskaya St., 3, 142290 Puschino, Moscow Region, Russia; eglork@gmail.com (E.V.M.); kimerzent@gmail.com (A.L.K.)

**Keywords:** polyelectrolyte microcapsules, allapinin, encapsulation, drug delivery, prolonged release

## Abstract

Allapinin has antiarrhythmic activity and can be used to prevent and treat various supraventricular and ventricular arrhythmias. Nevertheless, it is highly toxic and has a number of side effects associated with non-specific accumulation in various tissues. The complex of this substance with the monoammonium salt of glycyrrhizic acid (Al:MASGA) has less toxicity and improved antiarrhythmic activity. However, the encapsulation of Al:MASGA in polyelectrolyte microcapsules (PMC) for prolonged release will reduce the residual adverse effects of this drug. In this work, the possibility of encapsulating the allapinin–MASGA complex in polyelectrolyte microcapsules based on polyallylamine and polystyrene sulfonate was investigated. The encapsulation methods of the allapinin–MASGA in polyelectrolyte microcapsules by adsorption and coprecipitation were compared. It was found that the coprecipitation method did not result in the encapsulation of Al:MASGA. The sorption method facilitated the encapsulation of up to 80% of the original substance content in solution in PMC. The release of the encapsulated substance was further investigated, and it was shown that the release of the encapsulated Al:MASGA was independent of the substance content in the capsules, but at pH 5, a two-fold decrease in the rate of drug release was observed.

## 1. Introduction

Supraventricular and ventricular arrhythmias have a high prevalence of 2.25/1000 people in all age groups, with an annual increase in incidence of 35/100,000 people [1,2,3]. Both supraventricular and ventricular arrhythmias not only cause uncomfortable symptoms such as shortness of breath, sweating, dizziness, etc., but can also be life-threatening, increasing the risk of stroke and causing or worsening heart failure, and thus they can be fatal [4,5]. To treat this type of disease, a number of abnormalities such as hypoxia, acidemia, electrolyte disturbances, and hyperadrenergic state need to be addressed [6]. However, this goal may be difficult to achieve in the clinical circumstances in which this arrhythmia occurs. Slowing the rhythm with drugs that block the atrioventricular node may be more successful in the short term [6,7].

Allapinin (lappaconitine) is one such drug, which has antiarrhythmic activity through the inhibition of tetrodotoxin-sensitive, voltage-dependent sodium channels [8,9]. This substance is used in clinical practice in China and Russia for the treatment of supraventricular and ventricular arrhythmias [10,11]. However, this substance has a number of disadvantages: the high toxicity of the drug [12,13]; difficulty in the calculation of the therapeutic dose due to its cumulative accumulation in organs and tissues and non-linear pharmacokinetics [14]; having a short period for maintaining the therapeutic concentration of the drug in the blood [14]; and the occurrence of other side effects such as chills, shock, etc. [10]. One of the possible ways to eliminate the above disadvantages is the immobilization of allapinin for its prolonged release.

Wenxiu et al. demonstrated the possibility of immobilizing allapinin in the ionic biopolymer iota–carrageenan with a loading efficiency of 26.18% [15]. However, the total time of drug release from the iota–carrageenan complex in an aqueous medium at 37 °C and pH 7.4 was not more than 6 h, which may be insufficient because, according to the pharmacokinetic profile of the standard drug, its maximum concentration in the blood is maintained for 12 h after oral administration [11]. At the same time, lowering the pH of the incubation solution to 6.6 results in the complete release of the drug within 3 h of incubation, which may adversely affect the therapeutic effect of the drug in crossing the gastric acid barrier. Guo et al. proposed a more original approach: They combined allapinin with solid lipid nanoparticles (SLNs) and nanostructured lipid carriers (NLCs) and used transdermal administration by applying these parts to the skin surface [16]. This method of drug delivery made it possible to prolong the drug release time up to 70 h, but its concentration does not exceed 200 ng/mL, which is insufficient for a therapeutic effect. In other works devoted to the immobilization of allapinin, similar drawbacks are observed in the form of an insignificant increase in the drug release time or its low concentration in the solution [17,18,19]. Presumably, both drawbacks are related to the low solubility of allapinin [10].

The works of Uzbekov et al. proposed the encapsulation of a modified form of allapinin complexed with the monoammonium salt of glycyrrhizic acid (MASGA) [20]. The complex of this substance in a molar ratio of 1:4 with MASGA has a high solubility and more pronounced antiarrhythmic activity than simple allapinin and maintains the reduced contractile activity for a longer period [21,22]. In their study, the authors showed that when the encapsulated form of the allapinin–monoammonium salt of glycyrrhizic acid complex is released into the blood, the therapeutic dose is released over a longer period of time [20].

Another possible way to establish a prolonged system is to encapsulate the allapinin–monoammonium salt of glycyrrhizic acid antiarrhythmic complex in polyelectrolyte microcapsules (PMCs) based on polyallylamine and polystyrene sulfonate. Such PMCs are prepared by alternating the adsorption of oppositely charged polyelectrolytes on a micron-sized colloidal particle [23,24]. Our previous work showed the possibility of encapsulating an antiarrhythmic drug (amiodarone–MASGA) in polyelectrolyte microcapsules, with its prolonged release [25]. However, the efficacy of allapinin is higher than amiodarone [8,26,27], and the ability to control the size, the composition of the PMC shell, and its permeability at the stages of capsule preparation makes such a system more promising for solving the problems of antiarrhythmic therapy than existing analogs.

The aim of this work is to develop a system for the prolonged release of the allapinin–monoammonium salt of glycyrrhizic acid complex, by encapsulation in polyelectrolyte microcapsules obtained through the alternate adsorption of polystyrene sulfonate and polyallylamine on microsized CaCO_3_ cores.

## 2. Results and Discussion

In this work, the possibility of encapsulating the complex of allapinin–monoammonium salt of glycyrrhizic acid (1:4) (Al:MASGA) in polyelectrolyte microcapsules (PMCs) consisting of polystyrene sulfonate (PSS) and polyallylamine (PAH) was investigated. The effect of the components used to prepare the PMCs on the absorption peak of Al:MASGA at 255 nm was investigated.

### 2.1. The Effect of the Components (Salts and Polyelectrolytes) Used to Prepare the PMCs on the Absorption Peak of Al:MASGA

In the first step, the effect of CaCl_2_, Na_2_CO_3_, and NaCl salts used to form CaCO_3_ microspherulites and to remove the encapsulated drug residues in the solution was investigated. For this purpose, 50 μg/mL of the allapinin–monoammonium salt of glycyrrhizic acid complex was incubated with 0.33 M of CaCl_2_, 0.33 M of Na_2_CO_3_, or 0.5 M of NaCl for a maximum of 5 min. The results are shown in Figure 1.

As can be seen in Figure 1, the analyzed salts did not affect the absorption peak of Al:MASGA, suggesting that its structure and functional properties were preserved in the presence of these salts. From the obtained results, it can be concluded that these salts can be used in the encapsulation of the allapinin–monoammonium salt of glycyrrhizic acid complex.

The effect of the polyelectrolytes polystyrene sulfonate and polyallylamine on the absorbance peak of the complex of allapinin–monoammonium salt of glycyrrhizic acid at 255 nm was further investigated. For this purpose, 25 μg/mL of Al:MASGA was incubated with 1000 μg/mL of PSS or PAH, or in the presence of a 1:1 PSS-PAH complex for 72 h. The results are shown in Figure 2.

Figure 2 shows that the analyzed polyelectrolytes and their complexes did not affect the allapinin–monoammonium salt of glycyrrhizic acid complex absorption peak after 5 min of incubation. After 72 h of incubation, both polyelectrolytes reduced the Al:MASGA absorption peak: Polyallylamine reduced the absorption by 44%, and polystyrene sulfonate reduced it by 79%. However, the complex of these polyelectrolytes did not affect the absorption peak of the complex of allapinin–monoammonium salt of glycyrrhizic acid, probably due to the mutual compensation of the charged groups of these polyelectrolytes, resulting in a reduction in their interaction with Al:MASGA. It can therefore be concluded that these polyelectrolytes are suitable for the preparation of PMCs based on them since they are predominantly in a mutually compensated state in the shell composition.

### 2.2. Selection of a Method for Encapsulating the Al:MASGA

In the next stage of the work, we investigated the possibility of encapsulating the allapinin–monoammonium salt of glycyrrhizic acid (1:4) complex in polyelectrolyte microcapsules by coprecipitation. In the first step of the preparation of polyelectrolyte microcapsules, CaCO_3_ cores were formed in the presence of the target substance (Al:MASGA) with further coprecipitation, after which the supernatant was removed, and the absorbance of Al:MASGA at 251 nm was determined spectrophotometrically. As a result, it was shown that the use of the coprecipitation method in CaCO_3_ microspherulites did not allow for the incorporation of Al:MASGA into the spherulite, since the absorption peak at 251 nm after CaCO_3_ nucleation in the presence of Al:MASGA did not differ from the absorption peak of the allapinin–monoammonium salt of glycyrrhizic acid complex at the same dilution without the CaCO_3_ nucleation procedure.

In this context, it was proposed to incorporate the allapinin–monoammonium salt of glycyrrhizic acid complex into the spherulite through its sorption into CaCO_3_ nuclei. In this case, sodium carbonate microspherulites were formed separately and then incubated in an allapinin–monoammonium salt of glycyrrhizic acid solution for 1 h, after which the supernatant was collected, and the absorbance of Al:MASGA at 251 nm was determined spectrophotometrically. However, it was also found that using the sorption method in CaCO_3_ microspherulites did not allow for the incorporation of Al:MASGA into the spherulite. It can therefore be concluded that these methods are not suitable for the encapsulation of this substance.

In connection with the results described above, it was proposed to encapsulate the allapinin–MASGA complex using the adsorption method involving the sponge capsules of the composition (PSS/PAH)_3_, where PSS is the first layer, and the number “3” indicates three successive pairs of layers of PSS/PAH of the PMC shell. For this purpose, polyelectrolyte microcapsules were prepared through the layer-by-layer adsorption of the polyelectrolytes polystyrene sulfonate (PSS) and polyallylamine (PAH) onto the CaCO_3_ particle, followed by their dissolution by EDTA. A general scheme of the preparation of the polyelectrolyte microcapsules is shown in Figure 3.

The average diameter of the microcapsules was 4.5 μm, with a polydispersity index of 24.3%, and the ζ-potential of the microcapsule was +22 ± 3 mV.

Al:MASGA was incorporated into polyelectrolyte microcapsules using the adsorption method (concentration of Al:MASGA—330 μg/mL, incubation time—1 h). The result showed that approximately 75% of the substance was adsorbed.

### 2.3. Effect of Al:MASGA Concentration on the Dynamics of Incorporation and Release of Al:MASGA

The sorption method of the incorporation of a substance into PMCs may depend both on the time of the incubation of capsules in the solution with the substance and the concentration of this substance in the solution. Therefore, we studied the adsorption dynamics of the Al:MASGA PMC sponge structure (PAH/PSS)_3_ depending on the concentration of allapinin–MASGA: 330 μg/mL; 670 μg/mL; 1000 μg/mL; 1330 μg/mL; and 2000 μg/mL. The results are presented in Figure 4.

Figure 4A shows that the amount of substance adsorbed (in µg) increased with an increase in the concentration of Al:MASGA in the incubation solution. At the same time, the amount of adsorbed substance relative to its initial amount in the solution (in %) decreased with an increase in the concentration of the allapinin–MASGA complex in the incubation solution. At the same time, the figure shows that the incubation time had no effect on the amount adsorbed. Such an effect is probably due to the limited number of binding sites between Al:MASGA and PMC polyelectrolytes.

The dynamics of allapinin–monoammonium salt of glycyrrhizic acid complex release from the PMC of (PSS/PAH)_3_ composition was then studied to evaluate the yield of the substance over time. For this purpose, the microcapsules of the PMC of the composition (PSS/PAH)_3_ were incubated in water for 24 h. The obtained results are shown in Figure 5.

Figure 5A shows that a greater amount (μg) of the allapinin–monoammonium salt of glycyrrhizic acid complex was released from PMCs that contained a greater amount of encapsulated drug, and there was also a greater rate of drug release from such capsules. However, the increase in the rate of release of Al:MASGA was not proportional to the amount of Al:MASGA that the PMCs contained prior to desorption. As shown previously in Figure 4A, the capsules incubated in the 1330 μg/mL allapinin–monoammonium salt of glycyrrhizic acid complex solution adsorbed approximately 1150 μg of the substance, which was almost three-fold greater than the 330 μg/mL incubation, which led to the adsorption of approximately 380 μg. However, the rate of drug release from these capsules differed by less than two-fold. In addition, Figure 5B shows that the percentage of substance released (the amount of substance released relative to the amount encapsulated in PMC) decreased with the increasing amount of the allapinin–monoammonium salt of glycyrrhizic acid complex encapsulated. At the same time, the amount released did not exceed 50% after 24 h in the case of PMCs incubated in the Al:MASGA solution at concentrations of 670 µg/mL and 1330 µg/mL during the sorption step. It can therefore be concluded that increasing the amount of the encapsulated allapinin–monoammonium salt of glycyrrhizic acid complex leads to an increase in the time of its release from PMCs and a decrease in the dynamics of this release.

### 2.4. Effect of pH Medium on the Amount of Incorporated/Released Substance

Assuming that the interaction of the complex of allapinin–monoammonium salt of glycyrrhizic acid and polyelectrolyte microcapsules is of electrostatic nature, it was proposed to sorb Al:MASGA at different pH values, which may affect this interaction. The first step was to study the effect of the acidity of the solution on the absorption peak of Al:MASGA at 255 nm. For this purpose, the allapinin–monoammonium salt of glycyrrhizic acid complex was incubated in solutions with a pH of 3, 5, or 7. The obtained result is shown in Figure 6.

Figure 6 shows that the presented acidities of the solutions did not significantly affect the absorption peak of Al:MASGA, suggesting that the sorption of the allapinin–MASGA complex for its incorporation into polyelectrolyte microcapsules can be carried out at these pH values.

The next step was to determine the dynamics of sorption of the allapinin–monoammonium salt of glycyrrhizic acid complex in PMCs as a function of the pH of the incubation solution. For this purpose, the PMCs of the composition (PSS/PAH)_3_ were incubated in allapinin–MASGA complex (1000 μg/mL, 1.5 mL) solutions with pH 3, 5, or 7. The obtained results are shown in Figure 7.

As can be seen in Figure 7, the amount of the sorbed allapinin–monoammonium salt of glycyrrhizic acid complex did not differ regardless of the pH of the incubation solution.

Furthermore, the dynamics of the allapinin–monoammonium salt of glycyrrhizic acid complex yield from the PMC of the (PSS/PAH)_3_ composition was studied as a function of the pH of the incubation solution. For this purpose, the PMCs of the (PSS/PAH)_3_ composition were incubated in aqueous solutions with pH equal to 3, 5, or 7. The obtained results are shown in Figure 8.

Figure 8 shows that the amount of the allapinin–monoammonium salt of glycyrrhizic acid complex released did not differ at pH 3 and 7. However, at the pH of the solution equal to 5, there was an almost two-fold decrease in the dynamics of substance release from the PMC. The decrease in the dynamics of drug release at pH 5 will allow to an increase in the prolongation of drug release at its oral administration. It is known that the intraluminal pH of the gastrointestinal tract changes rapidly from very acidic in the stomach to about pH 6 in the duodenum. Subsequently, the pH in the small intestine gradually increases from pH 6 to about pH 7.4 in the terminal ileum [28]. Therefore, it is likely that changing the pH of the incubation medium can increase the time of substance release.

Thus, based on the results of this work, a method for the sorption of Al:MASGA in the PMC of the composition (PSS/PAH)_3_ was proposed, within which the possibility of encapsulating up to 80% of the initial content of the substance in the solution was demonstrated. The release dynamics of the complex of allapinin–monoammonium salt of glycyrrhizic acid were studied as a function of its initial content and the pH of the solution. The release of the encapsulated allapinin–monoammonium salt of glycyrrhizic acid complex was found to be independent of the solution pH and capsule content. 

## 3. Materials and Methods

### 3.1. Materials

The polyelectrolytes sodium polystyrene sulfonate (PSS) and polyallylamine hydrochloride (PAH), both with a molecular weight of 70 kDa, were purchased from Sigma (St. Louis, MO, USA). EDTA, Na_2_CO_3_, CaCl_2_, MgSO_4_, KH_2_PO_4_, KCl, and NaHCO_3_ were purchased from Reahim (Moscow, Moscow Region, Russian Federation). Supramolecular complex Al:MASGA (1:4) was prepared using MASGA (95% pure, Sigma-Aldrich, CAS No. 53956-04-0) and Allapinin^®^ (lappaconitine hydrobromide) (S. Yu. Yunusov Institute of the Chemistry of Plant Substances, Academy of Sciences of the Republic of Uzbekistan, Tashkent city, Uzbekistan).

### 3.2. Incorporation of Allapinin into Calcium Carbonate by Coprecipitation

Under stirring, 1.5 mL of a water solution of 0.33 M Na_2_CO_3_ was added to 1.5 mL of a water solution of 0.33 M CaCl_2_ with 660 μg/mL allapinin–MASGA (1:4) at 22 °C. The stirring time was 30 s. The suspension was incubated until the complete precipitation of the formed particles. The process of “maturation” of the microspherulites was monitored using light microscopy (determination of particle shape). The supernatant was then removed, and the precipitate was washed with water. Microparticles with an average diameter of 4.5 ± 1 μm were obtained. The size and number of microparticles were measured using the dynamic light scattering method on a Zetasizer nano ZS (Malvern, UK).

### 3.3. Incorporation of Allapinin into Calcium Carbonate Using the Adsorption Method

While stirring, 1.5 mL of a water solution of 0.33 M Na_2_CO_3_ was added to 1.5 mL of a water solution of 0.33 M CaCl_2_ at 22 °C. The stirring time was 30 s. The suspension was incubated until the complete precipitation of the formed particles. The process of “maturation” of the microspherulites was monitored via light microscopy (determination of particle shape). The supernatant was then removed, and the precipitate was washed with water. The obtained microspherulites were incubated in 330 μg/mL of the allapinin–MASGA complex solution for 1 h (pH—5, 22 °C). The supernatant was then removed, and the precipitate was washed with water. Microparticles with an average diameter of 4.5 ± 1 μm were obtained. The size and number of microparticles were measured using dynamic light scattering on a Zetasizer nano ZS (Malvern, UK).

### 3.4. Encapsulation of Allapinin in Polyelectrolyte Microcapsules with Dissolved CaCO_3_ Core (PSS/PAH)_3_

Polyelectrolyte microcapsules were prepared through the layer-by-layer adsorption of negatively and positively charged polyelectrolytes onto CaCO_3_ microspherulites, followed by the dissolution of CaCO_3_. While stirring, 1.5 mL of a water solution of 0.33 M Na_2_CO_3_ was added to 1.5 mL of a water solution of 0.33 M CaCl_2_ at 22 °C. The stirring time was 30 s. Polyallylamine (PAH) and sodium polystyrene sulfonate (PSS) layers were deposited on the surface of CaCO_3_ microspherulites from polyelectrolyte solutions (1.8 mL of solution contained 2 mg/mL of polyelectrolyte and 0.5 M NaCl). After each adsorption, CaCO_3_ particles with adsorbed polyelectrolytes were washed three times with 0.5 M of NaCl solution as necessary to remove the unadsorbed polymer molecules. The polyelectrolytes PSS and PAH were alternately adsorbed 3 times, resulting in 6-layer polyelectrolyte capsules containing a CaCO_3_ core. In the final step, this core was dissolved by incubating the capsule suspension obtained in 6 mL of 0.2 M EDTA solution. The particles were separated from the superfluid layer by centrifugation. The resulting capsules were washed three times with water to remove core degradation products. This procedure yielded the sponge capsules of the composition (PSS/PAH)_3_, where PSS is the first layer and the number “3” is 3 successive pairs of layers of PSS/PAH of the PMC shell. The resulting polyelectrolyte microcapsules were incubated in 1000 µg/mL of the allapinin–monoammonium salt of glycyrrhizic acid complex solution (1.5 mL) for 1 h at 22 °C, pH 5. (The incubation time, pH, and concentration of Al:MASGA may vary depending on the experiment and are additionally described in the Results section). After incubation, the supernatant was collected, and the absorbance of the Al:MASGA complex at 251 nm was determined spectrophotometrically. The supernatant was then removed, and the precipitate was washed with water. Microcapsules with an average diameter of 4.5 ± 1 μm were obtained. The size, number, and ζ-potential of microcapsules were measured using the dynamic light scattering method on a Zetasizer nano ZS device (Malvern, UK). 

### 3.5. Determination of Allapinin Release from PMCs

The PMCs were incubated in the solution for a long time until the release of the substance into the solution stopped. (The release time depends on the experiment described in the Results section.) At the same time, the concentration of the allapinin–monoammonium salt of glycyrrhizic acid complex was determined after each time period according to the experiment. PMCs were precipitated, and the supernatant was taken to determine the concentration of the allapinin–monoammonium salt of glycyrrhizic acid complex from the absorption peak at 255 nm using the spectrophotometric method. The supernatant was replaced after sampling.

### 3.6. Preparation of the 1:4 Complex of Allapinin and MASGA

A solution of MASGA (357.6 mg, MM894) in a 35% solution (100 mL) of EtOH in H_2_O was treated with a solution of lappaconitine hydrobromide (allapinin) (66.5 mg, MM 665) in EtOH (10 mL). The mixture was stirred on a magnetic stirrer for 1 h and evaporated in a rotary evaporator. The solid was freeze-dried to yield the 1:4 complex (434.1 mg) of allapinin and MASGA as a white amorphous powder.

### 3.7. Statistical Processing

Each experiment was replicated three times. Each sample was measured five times, and the values were calculated as mean and standard deviation. The significance of differences was tested using an independent two-sample *t*-test (Student’s *t*-test), *p* ≤ 0.01. To determine the concentration of the substance in relation to the optical density of the solution at 251 nm, a calibration curve was constructed using the equation y = 0.0103x + 0.0273 with a standard error of 0.00946.

## 4. Conclusions

In the framework of this work, the possibility of the encapsulation of the allapinin–monoammonium salt of glycyrrhizic acid (1:4) (Al:MASGA) complex into polyelectrolyte microcapsules (PMCs) consisting of polystyrene sulfonate (PSS) and polyallylamine (PAH) was studied. 

The first step was to investigate the effect of the components used in the preparation of PMCs on the absorption peak of Al:MASGA at 255 nm. It was shown that CaCl_2_, Na_2_CO_3_, and NaCl salts did not affect the absorption peak of Al:MASGA, whereas polyallylamine reduced the absorption by 44%, and polystyrene sulfonate reduced it by 79%. However, the complex of these polyelectrolytes did not affect the absorption peak of Al:MASGA, which allowed us to use these polyelectrolytes and the presented salts for the encapsulation of Al:MASGA.

In the next stage of the work, it was shown that the methods of the encapsulation of Al:MASGA through the coprecipitation and sorption of Al:MASGA into CaCO_3_ core were not suitable. However, the method of adsorption in capsules with a sponge structure of the composition (PSS/PAH)_3_ allowed 75% of Al:MASGA to be adsorbed from the solution at a concentration of 330 µg/mL. By studying the dynamics of the sorption of the allapinin–monoammonium salt of glycyrrhizic acid complex into polyelectrolyte microcapsules as a function of the concentration of this substance in the solution, it was found that the incubation time did not affect the amount of sorbed substance, with an increase in the concentration of the substance decreasing the % incorporated from the initial amount in the solution but also increasing the total amount of Al:MASGA incorporated into the PMC. The release of the sorbed substance from these polyelectrolyte microcapsules was further investigated, and it was shown that increasing the amount of the allapinin–monoammonium salt of glycyrrhizic acid complex encapsulated led to an increase in the time for its release from the PMC and a decrease in the dynamics of this release. 

The influence of the pH of the medium on the sorption and desorption of the allapinin–monoammonium salt of glycyrrhizic acid complex in polyelectrolyte microcapsules was further investigated. To this end, the effect of solution pH on the absorption spectrum of Al:MASGA at 255 nm was first determined. It was shown that the pH of the solution did not affect the absorption spectrum of Al:MASGA and that the pH of the incubation solution had no effect on the sorption of the substance into the polyelectrolyte microcapsules, but it had an effect on desorption. The amount of the allapinin–monoammonium salt of glycyrrhizic acid complex released did not differ at pH 3 and 7. However, at a solution pH of 5, an almost two-fold decrease in the rate of release from the PMCs was observed.

In the future, the form of the Al:MASGA complex encapsulated in polyelectrolyte microcapsules may be used orally for the gradual release of the substance in the intestine. Of particular importance is the gradual release of the substance over a period of more than 24 h, which may make it possible to reduce the number of drug intakes by the patient, increase the time needed for maintaining the therapeutic concentration of the drug in the blood, and achieve more linear pharmacokinetics. In particular, the effect of an increase in the release time of the substance at a pH close to the physiological pH in the small intestine should be taken into account.

## Figures and Tables

**Figure 1 ijms-25-02652-f001:**
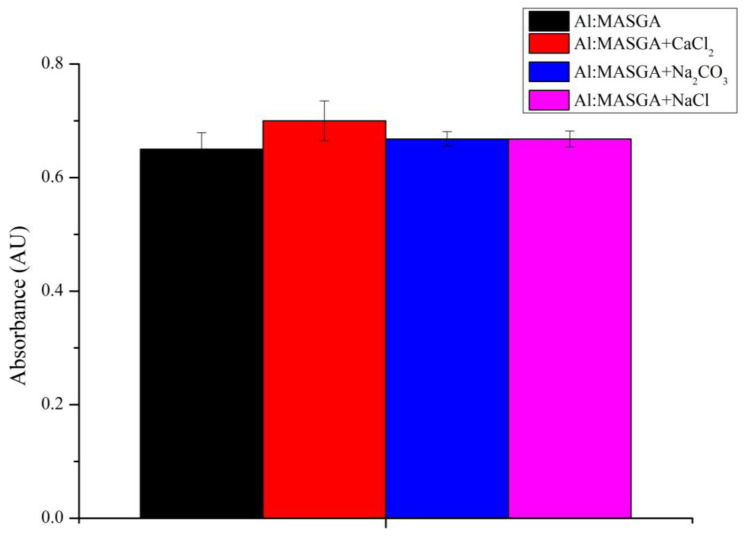
Absorbance of Al:MASGA at 255 nm in the presence of CaCl_2_, Na_2_CO_3_, or NaCl.

**Figure 2 ijms-25-02652-f002:**
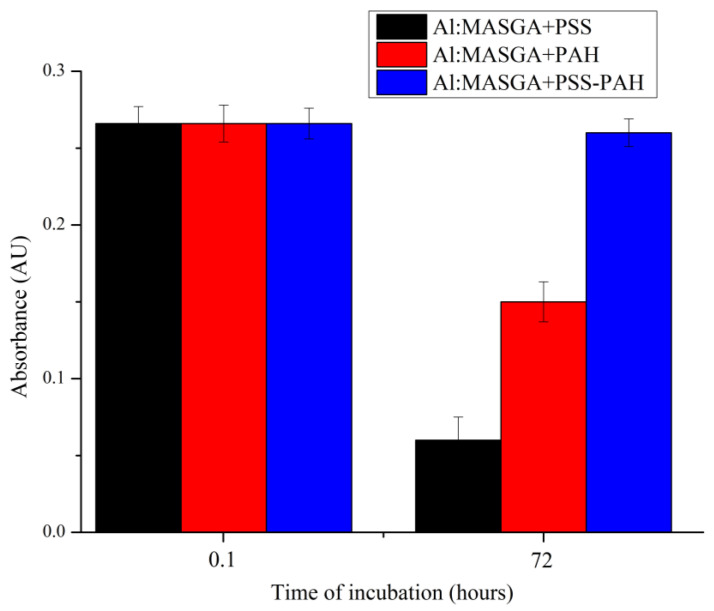
Absorbance of Al:MASGA at 255 nm when incubated with PSS, PAH, or PAH-PSS complex.

**Figure 3 ijms-25-02652-f003:**
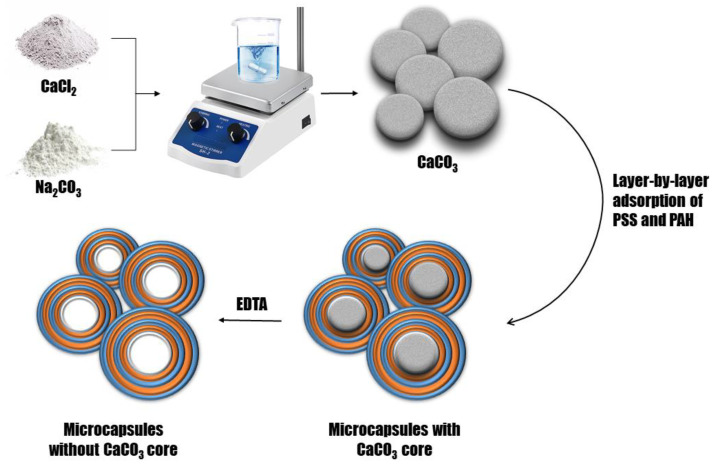
Scheme of preparation of polyelectrolyte microcapsules.

**Figure 4 ijms-25-02652-f004:**
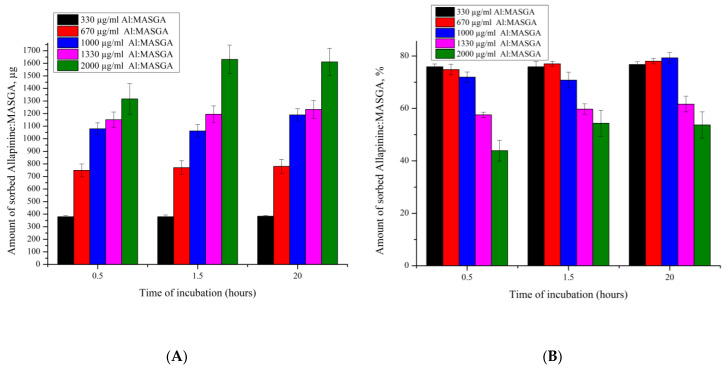
Amount of adsorbed Al:MASGA in PMC of composition (PSS/PAH)3 as a function of incubation time (pH—5, 22 °C): (**A**) amount of adsorbed substance, µg; (**B**) amount of adsorbed substance, %.

**Figure 5 ijms-25-02652-f005:**
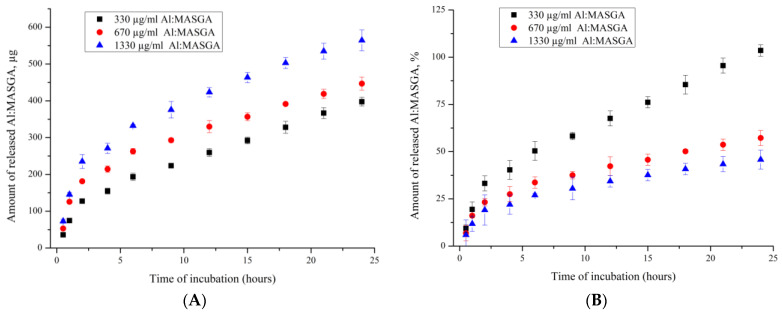
Dynamics of Al:MASGA yield from PMC of composition (PSS/PAH)_3_ (pH—5, 22 °C): (**A**) amount of desorbed material, µg; (**B**) amount of desorbed material, %.

**Figure 6 ijms-25-02652-f006:**
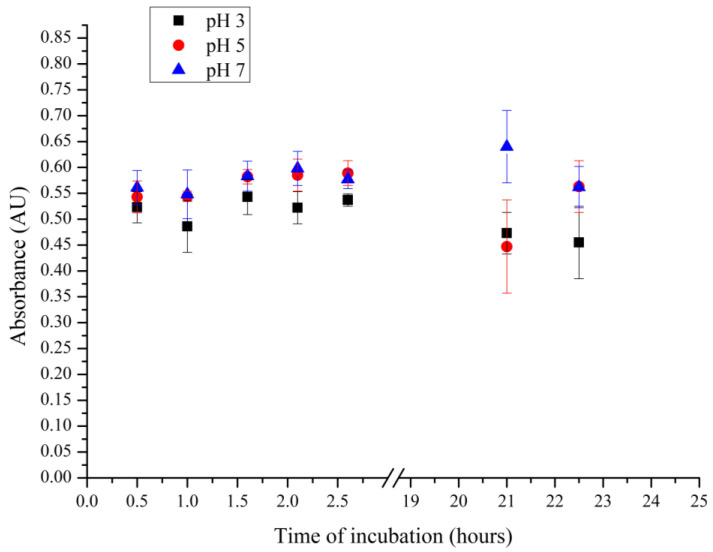
Absorbance of Al:MASGA at 255 nm in solutions of different acidities.

**Figure 7 ijms-25-02652-f007:**
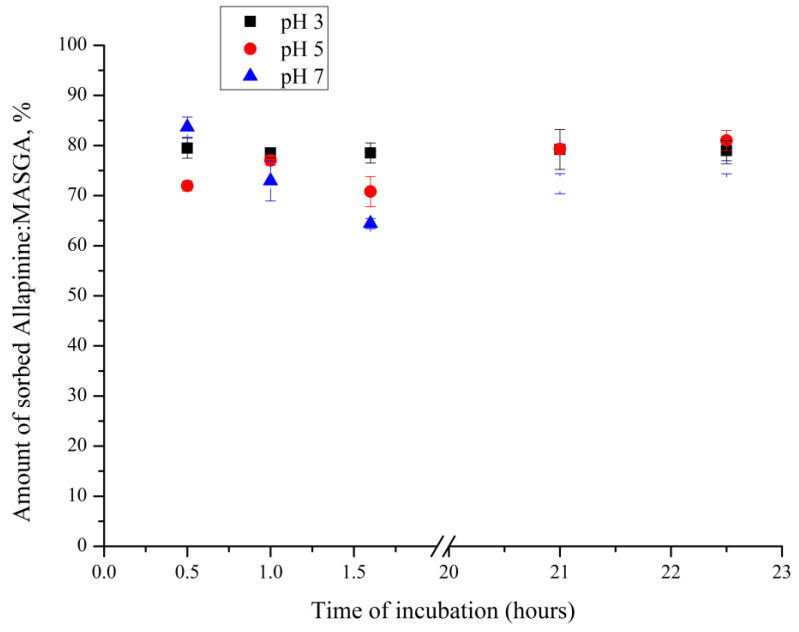
Amount of Al:MASGA adsorbed in the PMC of the composition (PSS/PAH)_3_ as a function of the pH of the solution (22 °C).

**Figure 8 ijms-25-02652-f008:**
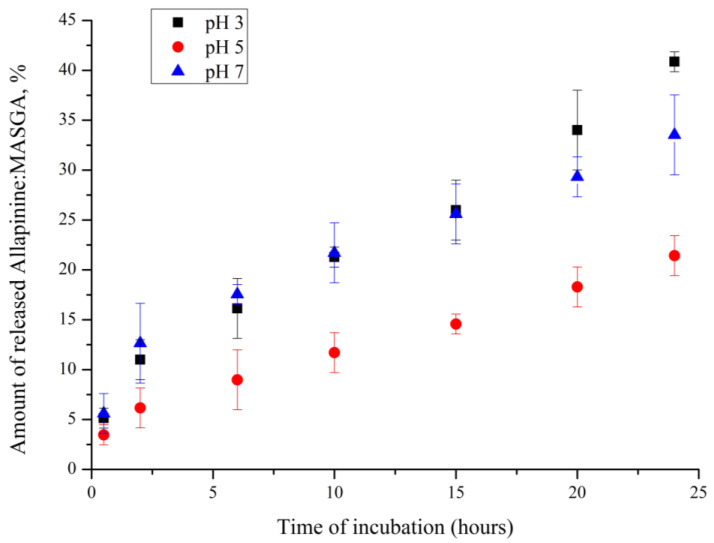
Dynamics of Al:MASGA yield from PMC of composition (PSS/PAA)_3_ as a function of solution pH (22 °C).

## Data Availability

Data are contained within the article.

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
