# Peer review of "Polyelectrolyte Microcapsules: An Extended Release System for the Antiarrhythmic Complex of Allapinin with Glycyrrhizic Acid Salt"

_ijms, 2024, doi:10.3390/ijms25052652_

Round 1
Reviewer 1 Report
Comments and Suggestions for Authors
In this draft, the author successfully highlights the necessity of the topic, specifically the prevalence of arrhythmia symptoms on the ventricles. The limitations of using allapinin in treatment and potential remedies are also discussed. However, the author only briefly introduces their previous research without delving into the specifics of the current study. Therefore, there is a need for the author to elaborate on the objectives and methods of the present research in the final part of the introduction. Regarding Figures 1, 2, and 6, the author should adjust the units of absorbance to adhere to scientific conventions. In the conclusion section, the author is advised to be more concise. Additionally, the author should emphasize the novel findings or significance of the current study and discuss how the overall research can be applied in practical scenarios. Comments on the Quality of English Language
Decimal numbers in the figures should be written according to English language principles.
Author Response
Thank you for your attention. The corrections have been made to the article. We appreciate your participation in the review process.
- At the end of the introduction, the aim of the manuscript and the methods to achieve the aim were added.
- Thank you for bringing this to my attention. We have reviewed the figures and made the necessary adjustments to ensure that the units of absorbance adhere to scientific conventions.
- Thank you for your feedback. We appreciate your suggestion to be more concise in the "Conclusion" section. We have taken your advice into consideration and made the necessary revisions. Furthermore, We have emphasized the novelty and significance of my research, as well as discussed its potential practical applications. We believe these additions will enhance the overall impact of the study.
Reviewer 2 Report
Comments and Suggestions for Authors
Shavkat I. Salikhov et al., in the manuscript “Polyelectrolyte Microcapsules - Extended Release System for the Antiarrhythmic Complex of Allapinin with Glycyrrhizic 3 Acid Salt “ describe the possibility of encapsulation of allapinin: mono-ammonium salt of glycyrrhizic acid (1:4) (Al:MASGA) complex into polyelectrolyte microcapsules (PMCs) consisting of polystyrene sulfonate (PSS) and polyallylamine (PAH) was studied. Allapinin has antiarrhythmic effects and can be used to prevent and treat various supraventricular and ventricular arrhythmias. Nevertheless, it is highly toxic and has a number of side effects related to non-specific accumulation in various tissues. Therefore, it seems justified to search for new formulations that will reduce side effects.
While this study presents several interesting insights, the manuscript requires substantial revision on writing. There are numerous formatting issues, and the writing is difficult to understand.
Introduction meets the requirements. The literature included in this section is current.
The results should include subsections to make it easier for the reader to read.
Figure 1, the y-axis should end at 0.8. Decimal values should be separated from the whole by a dot, not a comma. Instead of 0,6 it should be 0.6. The x axis description is missing. What is the error size represented by the error bars on the chart?
Figure 2, the y-axis should end at 0.8. Decimal values should be separated from the whole by a dot, not a comma. Instead of 0.6 it should be 0.6. What is the error size represented by the error bars on the chart?
The results contain data that should be placed in methods, e.g. Line 91-93, Figure 3
Figure 4 -8 What is the error size represented by the error bars on the chart?
The article shows that the results come with a discussion, so the chapter should be called 'Results and discussion'.
The methods do not have a statistics subchapter, which would provide information on how many repetitions were used and what are the standard deviations? Line 245 - please provide the city, state, country where you purchased the reagents, e.g. KCl.
There is no table showing the composition of all formulations and the research methods used for them.
Conclusion
The conclusion part is too long. There is no explanation of what significance these studies have for the entire research discipline.
Author Response
Thank you for your questions. We appreciate your feedback and we are glad to hear that you found our article interesting.
- We are grateful for your response.
- Subsections were added for better understanding of material.
3-4. Thank you for your attention, the correction has been made in the article.
Also, we corrected separation of decimal values. The x axis is not missing, we try to visualize the absorbance of allapinin in a presence of different salts, the name of each sample was indicated in a legend of figures. The error size of the error bar was calculated as a standard deviation from average value based on five time measurement of absorbance of sample.
- We agree with that comment and appreciate for that. Result section was corrected, some of the methods was transferred to materials and methods section.
- Figure 4 -8. The error size of the error bar was calculated as a standard deviation from average value based on measurements of absorbance of five duplicated samples.
- We changed the name of chapter.
- We added subchapter about Statistical Processing.
- Since the experiments with calcium carbonate (sorption and coprecipitation) showed zero values, and the inclusion of the substance mteodom sorption with different concentrations of sorbate, and the experiment on the effect of ph was carried out with one concentration of sorbate, bringing these data into one table will confuse the reader and make it much more difficult to understand. Therefore, the authors believe that the inclusion of a table in this article is unnecessary.
10. We make conclusion shorter and added explanation part.
Reviewer 3 Report
Comments and Suggestions for Authors
I recommend the rejection information because there is not adequate information in the experimental section for a proper review see comment 3.
1. Page 1 Line 21: The abbreviation PMC should be put in parenthesis at line 15 where “polyelectrolyte microcapsules” were first used.
2. Page 9 lines 243-244 The molecular weight of sodium polystyrene sulfonate is missing. The providers of allapinin and the mono ammonium salt of
glycyrrhizic acid, are also missing.
3. Page 9 lines 248-249 “Under stirring, 0.33 M Na2CO3, 0.33 M CaCl2 and allapinin: monoammonium salt of glycyrrhizic acid complex were added.” Assuming the authors refer to aqueous solutions of Na2CO3, and CaCl2 six questions arise: 1) The quantities of the solutions. 2) The quantity of the complex 3) The ratio of allapinin: monoammonium salt of glycyrrhizic acid (this is revealed in the conclusions) 4) The preparation method of the complex 5) The complex was added in what form solution? Powder? 6) Where all these were added. The same tactic is used throughout the entire experimental section. For example, lines 284-285 “The PMCs were incubated in solution for a long time until the release of the substance
into the solution stopped.” How much? time? pH? Temperature? Critical information is missing
The quality of English is not the major problem of the manuscript
Author Response
We thank the reviewer for leaving comments on our paper. We realize that there are still questions about the materials and method of our paper. Therefore, the materials and methods section has been significantly revised and all the reviewer's comments have been taken into account. Also, to improve the understanding of the material, explanations of the experimental conditions have been added and are reflected in the Results and Discussion and Materials and Methods sections.
Reviewer 4 Report
Comments and Suggestions for Authors
I had the opportunity to review the manuscript entitled “Polyelectrolyte Microcapsules - Extended Release System for the Antiarrhythmic Complex of Allapinin with Glycyrrhizic Acid Salt” submitted to Int. J. Mol. Sci.
I agree with its publication in Int. J. Mol. Sci., but only after some necessary changes are taken care of:
1) The authors mention a repetition of the full name in allapinin: monoammonium salt of glycyrrhizic acid complex throughout the whole text. Since they have introduced at the beginning of the manuscript the abbreviation Al:MASGA, they could as well adopt it further on and replace the full name with the abbreviated term.
2) Explain what number 3 means in the name (PSS/PAH)3. Is it that the authors created capsules with 3 successive layers of PSS/PAH complexes? It should be clarified and better described in the experimental part.
3) In Figure 3 it is presented that EDTA was used to remove CaCO3. If this is correct, it should be mentioned in the experimental part (Section 3.4) and also mention EDTA in the materials.
4) In Section 3.5: For the determination of allaphin release from PMCs: Was a calibration curve created concerning the determination of the allapinan:monoammonium salt of glycyrrhizic acid complex concentration? The authors should provide more info on that. For example, what solutions of allapinan:monoammonium salt of glycyrrhizic acid complex were prepared (in which solvent, at what pH solution value, etc.).
5) In Figure 8, dynamics of Al:MASGA yield from PMC of composition (PSS/PAA)3 as a function of solution pH is presented. What system of Al:MASGA-loaded PMC was used? I mean which was the initial content of drug in the PMCs (one of these: 330, 670 or 1330 μg/ml, like in Figure 5?)? And is this Figure 8 somehow compared to the findings of Figure 5? What was the pH of the experiment solutions in Figure 5?
6) The authors should provide a discussion, comparing their new results to the ones previously published in [25] and evaluate the performance of their new system.
Author Response
Our group would like to thank the reviewer for leaving a detailed feedback that helped us to improve the quality of our work significantly. We were happy to answer all the reviewer's questions:
- We agreed with reviewer and changed most part of repetition of the full name for abbreviation.
- The number "3" is 3 successive pairs of layers of PSS/PAH of the PMC shell. We added the better explanation in the text.
- Thank you. We missed that. Description processes with EDTA added to the materials section.
- Added more information about that in materials and method section.
- The initial drug content was 1000 µg/mL 1.5ml, this observation is accepted and has been taken into account in the results and added to the materials and methods.
It is not possible to compare directly, but in general this does not affect the conclusions obtained, since the purpose of these experiments was to determine whether there is an effect of pH and concentration of the substance on its incorporation into PMC.
The pH in figure 5 is 5. Added this to the results and materials and methods.
- The previous work used Amiodarone-MASGA (1:8) as the active ingredient and this work uses Allapinin-MASGA (1:4). These studies cannot be compared because amiodarone and allapinin differ in a number of parameters. They have different structure and chemical composition, and therefore their binding mechanisms with PMC may differ. In addition, according to literature data, the oral therapeutic dose of amiodarone ranges from 800 to 1600 mg (https://www.ncbi.nlm.nih.gov/books/NBK482154/), while allapinin is 75-150 mg (https://pubmed.ncbi.nlm.nih.gov/24881307/, PMID: 2478747). Also, these drugs have different levels of toxicity.
Round 2
Reviewer 3 Report
Comments and Suggestions for Authors
The authors improved substatially the experimental section. By the way Subchapter 3.7 Preparation of the 1:4 complex of allapinin and MASGA sould be numbered 3.6.
Comments on the Quality of English LanguageMinor corrections are needed
Author Response
Thank you for your review. We have made the necessary adjustments to the numbering within the Materials and Methods section of our article.
Reviewer 4 Report
Comments and Suggestions for Authors
I thank the authors for taking into account my comments/suggestions.
I have no further comments to make, to me the article can be published at its present form.
Author Response
Thank you for your prompt review. We greatly appreciate your acknowledgment of our efforts to address your comments and suggestions.
We are grateful for your time and expertise in evaluating our work.